# FuncPEP v2.0: An Updated Database of Functional Short Peptides Translated from Non-Coding RNAs

**DOI:** 10.3390/ncrna10020020

**Published:** 2024-04-09

**Authors:** Swati Mohapatra, Anik Banerjee, Paola Rausseo, Mihnea P. Dragomir, Ganiraju C. Manyam, Bradley M. Broom, George A. Calin

**Affiliations:** 1Department of Translational Molecular Pathology, University of Texas MD Anderson Cancer Center, Houston, TX 77030, USA; smohapatra@mdanderson.org (S.M.); rausseopaola@gmail.com (P.R.); 2The University of Texas MD Anderson Cancer Center UTHealth Houston Graduate School of Biomedical Sciences, Houston, TX 77030, USA; anik.banerjee@uth.tmc.edu; 3Department of Neurology, University of Texas McGovern Medical School, Houston, TX 77030, USA; 4Scripps College, Claremont, CA 91711, USA; 5Institute of Pathology, Charité—Universitätsmedizin Berlin, Corporate Member of Freie Universität Berlin and Humboldt-Universität zu Berlin, 10117 Berlin, Germany; mihnea.p.dragomir@gmail.com; 6German Cancer Consortium (DKTK), German Cancer Research Center (DKFZ), 69120 Heidelberg, Germany; 7Berlin Institute of Health at Charité, 10117 Berlin, Germany; 8Department of Bioinformatics and Computational Biology, The University of Texas MD Anderson Cancer Center, Houston, TX 77030, USA; gcmanyam@mdanderson.org (G.C.M.);; 9Center for RNA Interference and Non-Coding RNAs, The University of Texas MD Anderson Cancer Center, Houston, TX 77030, USA

**Keywords:** non-coding RNAs, ncRNA-encoded peptides, immunity, micro-peptides

## Abstract

Over the past decade, there have been reports of short novel functional peptides (less than 100 aa in length) translated from so-called non-coding RNAs (ncRNAs) that have been characterized using mass spectrometry (MS) and large-scale proteomics studies. Therefore, understanding the bivalent functions of some ncRNAs as transcripts that encode both functional RNAs and short peptides, which we named ncPEPs, will deepen our understanding of biology and disease. In 2020, we published the first database of functional peptides translated from non-coding RNAs—FuncPEP. Herein, we have performed an update including the newly published ncPEPs from the last 3 years along with the categorization of host ncRNAs. FuncPEP v2.0 contains 152 functional ncPEPs, out of which 40 are novel entries. A PubMed search from August 2020 to July 2023 incorporating specific keywords was performed and screened for publications reporting validated functional peptides derived from ncRNAs. We did not observe a significant increase in newly discovered functional ncPEPs, but a steady increase. The novel identified ncPEPs included in the database were characterized by a wide array of molecular and physiological parameters (i.e., types of host ncRNA, species distribution, chromosomal density, distribution of ncRNA length, identification methods, molecular weight, and functional distribution across humans and other species). We consider that, despite the fact that MS can now easily identify ncPEPs, there still are important limitations in proving their functionality.

## 1. Introduction

Conventional studies have defined non-coding RNAs (ncRNAs) as having no protein-coding potential [1]. However, several ncRNAs with small open reading frames (smORFs) have been reported to be translated into short functional peptides, which, by definition, are less than 100 aa in length [2]. By comparison, more than 95% of the proteins translated from coding regions are significantly longer than 100 aa [3,4,5]. Peptides encoded by ncRNAs are termed ncRNA-encoded peptides (ncPEPs). ncPEPs have been extensively characterized in silico [6,7], but validation by wet lab experiments using high-throughput mass spectrometry (MS) and ribosome profiling is still lacking for many [8,9]. In addition, the functional characterization of ncPEPs has not been extensively studied compared to peptides derived from coding regions. smORFs in the transcripts of ncPEPs lack evolutionary conservation compared to the peptides derived from coding regions [9], leading to tissue-specific expression in certain species [1] and most likely in specific pathological states. ncPEPs can be used as biomarkers for tissue differentiation under physiological conditions and during disease progression. There was a spike in the number of discoveries and reports of FuncPEPs in 2020, the year the first version of a FuncPEP was published [10] and, therefore, we aim to keep the database updated to motivate researchers in the field to be on the lookout for novel functional peptides while they study the classical and nonclassical functions of ncRNAs [11,12].

Almost all classes of ncRNAs have been shown to encode ncPEPs. ncRNAs are broadly classified according to their length into long ncRNAs (lncRNAs), which are longer than 200 bp, and short non-coding RNAs (sncRNAs), which are shorter than 200 nucleotides (nts). Furthermore, different classes of lncRNAs are reported based on their genomic location, such as long intergenic ncRNAs (lincRNAs), or based on their conservation across species such as transcribed ultraconserved regions (T-UCRs), which are transcribed from ultraconserved genomic regions. Similarly, sncRNAs are classified into several categories, including microRNAs (miRNAs), PIWI-interacting RNAs (piRNAs), and transfer RNAs (tRNAs). Circular RNAs (circRNAs) or small nucleolar RNAs (snoRNAs) belong to both classes depending on their bp length. lncRNA and sncRNA have been extensively studied and reported to be involved in maintaining homeostasis [13] and contributing to pathological conditions [14,15]. The function of ncRNA transcripts is complex, and in recent years we have seen that most of these classes can also function by encoding ncPEPs, which broadens their mechanistic spectrum.

ncPEPs are mainly identified by computational prediction software for potential ORFs, and out of the many predicted ncPEPs, only a handful of them have been characterized by MS and techniques such as Western blotting, immunohistochemistry, and ribosome profiling. Moreover, the functional characterization of ncPEP is difficult and limited; thus, more effective techniques for functional validation are warranted to expand this field. Only a few experimentally discovered ncPEPs have been functionally characterized in the last decade with an aim to understand their role in ncRNA-associated pathology and in tissue homeostasis [16,17]. The biologically active short ncPEPs are functionally distinct from peptides translated from highly conserved messenger RNAs (mRNAs) [18] or short peptides generated through the breakdown of large proteins. For example, human smORFs are not always conserved beyond primates, while several human proteins are highly conserved across species [19,20]. Regarding the function of human ncPEPs, they are mostly involved in rapidly changing adaptive processes such as immunity and tumorigenesis [13,21].

Studies reporting ncPEPs using indirect (in silico analyses and peptide sequencing data without molecular validation) and predictive methods need more characterization and cannot be argued as being functional. Therefore, in 2020, we put together the first database [10] of **fu**nctional **nc**RNA-encoded **pep**tides (FuncPEPs) only including ncPEPs that had been validated by experimental methods and functionally characterized. As proteomics technologies have significantly developed, we hypothesized that, in recent years, the discovery of ncPEPs has escalated and, therefore, we performed an update of our initial database. This revised version of the database (FuncPEP v2.0) includes the new functional ncPEPs that have been published in the last 3 years along with the categorization of the host ncRNA from which they are derived.

## 2. Systematic Profiling, Data Collection, and Construction of Database

### 2.1. Collection of Data and Database Construction

The selected peptides were translated from ncRNAs containing smORFs and were included in the database as per the same criteria as in the first version of FuncPEP: (1) validated by molecular biology techniques (MS, Western blotting, immunohistochemistry, immunofluorescence) and/or indirect methods such as ribosome profiling and loss/gain of function studies by wet lab experiments; (2) functionally characterized by demonstrating their involvement in physiological homeostasis or disease; and (3) size less than 100 aa in length. Some of the ncPEPs are validated only by indirect methods and are marked with an asterisk in the database and require further validation. ncPEPs identified by computational methods only were not included in the database. We used the same inclusion criteria as in the first version of FuncPEP [10], performing searches of PubMed and Google Scholar to identify and characterize ncPEPs in the period from 2020 to present using keywords (“functional peptides”, “antisense RNA”, OR “lincRNA”, OR “lncRNA”, OR “miRNA”, OR “circRNA”, OR “rRNA”, OR “tRNA”, OR “ncPEP”). Furthermore, these terms were filtered by incorporatingterms regarding “translation” (“smORF”, OR “Ribo-seq”, OR “ribosome profiling”, OR “mass spectrometry”, OR “translation”).

### 2.2. Selection Criteria for ncPEPs Included in FuncPEP 

A PubMed search incorporating the keywords was performed and screened for validated functional peptides derived from non-coding RNAs. The novel identified ncPEPs included in our database were characterized by a wide array of molecular and physiological parameters (i.e., types of host ncRNA, species distribution, chromosomal density, distribution of ncRNA length, identification methods, molecular weight and functional distribution across humans and other species). Exclusive to our database, we only included ncPEPs that were detected by direct means (MS, Western blotting, immunohistochemistry, immunofluorescence) or indirect means (ribosome profiling, loss/gain of function studies) of experimental confirmation. All ncPEPs from the database are also functionally characterized and are linked with physiological and pathological processes as well.

## 3. Database Construction and Results

### 3.1. Systemic Review and Database Interface

A total of 15,805 articles were identified for the period August 2020 to July 2023. After accounting for duplicate articles, 10,502 articles were initially screened by title and abstract. Subsequently, 927 articles were selected for full-text evaluation to identify a wide range of biological and molecular features (i.e., species specificity, type of RNA source, amino acid length, etc.), which are easily accessible through FuncPEP’s user-friendly interface. Of these 927 candidate studies, only 40 studies, with a total of 40 validated functional ncPEPs, were included in the FuncPEP v2.0 database, according to the stated inclusion criteria (Figure 1A). The updated version of the website, described in this manuscript, includes the most recently investigated and characterized FuncPEPs to date (Figure 1B). Moreover, it was noted that there has not been a large increase in the number of ncPEPs identified in the last 3 years (Figure 1B). Throughout the last decade, the number of identified functional peptides has increased due to advances in standardized techniques.

FuncPEP is designed to have a better, user-friendly interface, allowing researchers to access all the validated ncPEPs discovered so far through the website. The website can be accessed via the following link: https://bioinformatics.mdanderson.org/Supplements/FuncPEP/ (accessed on 27 March 2024). The major sections of the website are divided into the (1) Home section, which includes a description of FuncPEP; (2) Database section, which includes a dynamic hyperlinked table browser providing detailed information about all the ncPEPs; (3) Method section, which includes a description of the workflow that we used to collect and curate the ncPEP characteristics; and (4) Help section, which includes details and aids the user in website navigation (Figure 2A–D).

### 3.2. Characteristics of ncPEPs from the FuncPEP v2.0 Database

FuncPEP v2.0 has been designed to provide a dynamic resource for accumulating ncPEP data from peer-reviewed scientific literature. Our database also provides information on the molecular characteristics of the included ncPEPs. The overall function of the database website contains information and characteristics of ncRNAs, as follows: the chromosomal position, ncRNA length, amino acid sequence, the method of identification, and the physiological function across all species. The analysis of the ncPEPs’ lengths in aa showed that the average length range of the included ncPEPs is 51 to 60 aa (Figure 3A). FuncPEP v2.0 also provides the molecular weight of the selected ncPEPs across all species, with an average range of 2 to 5.99 kDa (Figure 3B). Next, we sought to analyze the types of ncRNA that encode for ncPEPs. As expected, the majority of the ncPEPs are encoded by lncRNAs (69.4%, 107), followed by miRNAs (9.71%, 15), and circRNAs (3.89%, 6) (Figure 3C). Previous reports have shown that the majority of circRNAs and pri-miRNAs are also derived from lncRNAs [22]. In addition, FuncPEP v2.0 provides the host ncRNA length of the selected ncPEPs, across all species, with a range of 1000 to 10,000 bp (Figure 3D), indicating that lncRNAs have regions that may become spliced before the initiation of translation. Further investigation of the altered splicing frequencies in ncRNAs correlating with their biological function and structural characteristics are warranted. 

We then analyzed in which species the selected ncPEPs were discovered, of which, the majority were found in *Homo sapiens* (66.2%, 102), followed by *Mus musculus* (9.74%, 15), and *Drosophila melanogaster* (7.14%, 11) (Figure 3E). In addition, bioinformatically predicted smORFs located in ncRNA regions were experimentally confirmed to be translated. Advances in metagenomics and transcriptomic platforms have been shown to improve the accuracy of predicting the translation of smORFs to ncPEPs [23]. Therefore, bioinformatics tools for smORF prediction were included in the FuncPEP database. The majority of the experimental methods were ribosome profiling (42.8%, 66), followed by Western blotting (16.2%, 25), and MS (14.3%, 22) (Figure 3F). 

The ncPEPs’ chromosome distribution demonstrated that chromosomes 17, 19, and 20 had a higher number compared to the other chromosomes carrying the candidate ncPEPs (Figure 4). Interestingly, an analysis of the chromosomal distribution of the human ncPEPs demonstrated that chromosome 17 displayed the largest presence of ncPEPs. Furthermore, an excess of genes on chromosome 19 has also been reported for protein-coding genes [24]. This suggests that the higher the gene density a chromosome exhibits, the more likely that it has an encoding region for an ncPEP. Further supporting this assumption, we observed that the ncPEPs are found in all chromosomes except chromosome 13.

### 3.3. The Functions of ncPEPs

The selected ncPEPs in the FuncPEP database are observed to play cell-specific roles in regulating multiple cellular processes, including metabolism and immunity. The function of ncPEPs in tumor-promoting cellular pathways has been well recognized [25], more specifically, in myeloid malignancies [26]. Furthermore, biomarker discovery platforms have associated the higher expression of these ncPEPs with poor prognosis and survival rates [27]. To decipher the role of the retrieved funcPEPs, we performed a functional summary and meta-analysis of the candidate ncPEPs included in the v2.0 database across all species and then *Homo sapiens* separately (Figure 5A,B). Interestingly, immunity was the most enriched function across all species, including *Homo sapiens* (Figure 5A,B). Several of the included ncPEPs in the FuncPEP database predominantly regulated host pro- or anti-tumor immunity. Other top functions across all species were development (*n* = 27, 17.53%), tumor suppressor (*n* = 13, 8.44%), tumor-promoting (*n* = 12, 7.79%), metabolism (*n* = 6, 3.89%), and muscle contractility (*n* = 5, 3.24%) (Figure 5A). Interestingly, similar patterns of functionality were observed in the ncPEPs identified in *Homo sapiens*, the top five enriched functions being immunity (*n* = 63, 61.16%), tumor-promotion (*n* = 12, 11.65%), tumor suppression (*n* = 12, 11.65%), neural regeneration (*n* = 3, 2.91%), and metabolism (*n* = 3, 2.91%) (Figure 5B). This functional analysis reveals that these classes of molecules have a diverse set of functions in a wide array of regulatory cellular processes, which warrant future study. 

Further, Figure 5C highlights the most enriched pathways individually stratified based on the species. A comprehensive overview of the wide array of functions driven by the ncPEPs in the database revealed that across all species, the majority of the ncPEPs are involved in immunity (*n* = 69, 44.81%). Interestingly, the most common enriched function across each species separately was development, specifically for *Arabidopsis thaliana*, *Brassicacease*, *Drosophila melanogaster*, *Medicago truncatula*, *Physcomitrella patens*, and *Danio rerio*. 

## 4. Discussion

The FuncPEP v2.0 database is designed to be a comprehensive resource using experimentally confirmed ncPEPs translated from ncRNAs and not only by bioinformatic tools. The database has been updated from the previous version based on functionally and experimentally validated ncPEPs. Ideally, the use of prediction tools for the initial screening step is robust, but it is important to experimentally characterize and validate their expression, localization, abundance, and other biochemical properties. Based on the criteria set in our previous version, we maintain our decision to not include ncPEPs discovered using ribosome profiling without independent experimental validation in version 2.0. 

This observation could mean that (i) ncPEPs are not as abundant as originally predicted; or (ii) the current methodology for discovering, mapping, and validating ncPEPs is limited. The preliminary analysis of the candidate functional ncPEPs revealed that a wide range of species harbor a diverse and mechanistically versatile group of ncRNAs that are translated into functional small peptides. Although the functional relevance of these ncPEPs in biological settings has increasingly been recognized, the mechanistic underpinnings and cell-specific functions in different species are inconclusive due to the paucity of identification techniques. The clinical implications and discovery platforms for the identified ncPEPs are warranted for future investigation as a mode of patient stratification for disease severity and progression. More recently, multiple methods have been incorporated to comprehensively identify ncPEPs, primarily encompassing computational (i.e., intrinsic sequence features as a proxy of open reading frame (ORF) length, sequence homology to documented protein sequences, nucleotide composition, and substitution ratio used to putatively characterize the coding potential of ncRNA reading frames) [28,29,30] and experimental (i.e., ribosome profiling, mass spectrometry, and global translation initiation sequencing) [31,32] approaches for ncPEP classification. Understanding the biological roles of ncRNAs can provide a tool for examining the molecular mechanisms of understanding cell growth, proliferation, and development, as well as their responses to environmental stressors.

The downstream functional analysis of the peptides included in the FuncPEP v2.0 database revealed that the majority of the characterized ncPEPs play regulatory roles in immune function and development. For example, miPEP155 selectively binds to HSC70, rendering the modulation of MHC class II presentation in dendritic cells (DCs). Targeting miPEP155 in a murine model of autoinflammation exhibiting a dampening of DC-mediated autoimmunity [33] highlighted the importance of an ncPEP–immune axis, warranting future investigation. To gain translational insight, the authors also classified the potential of miPEP155 to interact with HSC70 in human DCs [33]. Fluorescent-tagged miPEP155 was observed to localize within the cytoplasmic and nuclear regions of human DC subsets. A large number of small peptides identified in our database regulate the function of antigen presentation [33], which could potentially provide an additional layer of refined immune regulation as well as potential peptide drug candidates for therapeutic intervention in a wide range of immunogenic diseases.

Studies investigating the functional roles of ncPEPs could help us understand their roles in cell-to-cell communication and delineate the complexity of ncRNAs with translational capabilities. The addition of several novel ncRNAs along with multiple regulatory roles has inspired researchers to investigate their multifaceted roles. However, with the discovery of novel functional peptides being translated from non-coding RNA, a new horizon has opened for researchers to investigate the function of non-coding regions at the genomic, transcriptomic, and proteomic levels. 

We plan to continue to regularly screen for newly discovered ncPEPs with functional roles and include them in the FuncPEP database. In addition, we encourage researchers to inform us about new ncPEPs for immediate inclusion in our database. This field has the potential to expand the world of ncRNA biology and translate ncRNAs into clinical practice as biomarkers, with uses such as the identification of circulating ncPEP plasma markers. New avenues could be opened, enabling a deeper understanding of multi-level peptide–peptide, peptide–DNA, and peptide–ncRNA interactions, which could lead to the discovery of new therapies.

## Figures and Tables

**Figure 1 ncrna-10-00020-f001:**
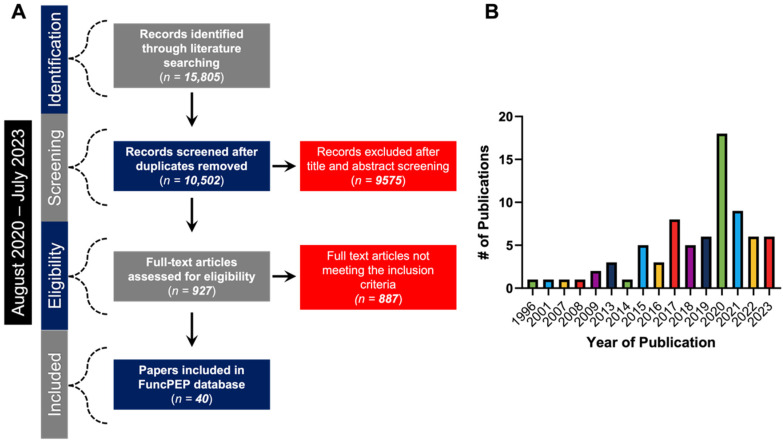
(**A**) Workflow demonstrating the selection criteria (i.e., identification, screening, eligibility testing, and inclusion) for FuncPEPs included in the database. (**B**) Analysis indicating the number of publications pertaining to functional ncPEPs. # means “Numbers”.

**Figure 2 ncrna-10-00020-f002:**
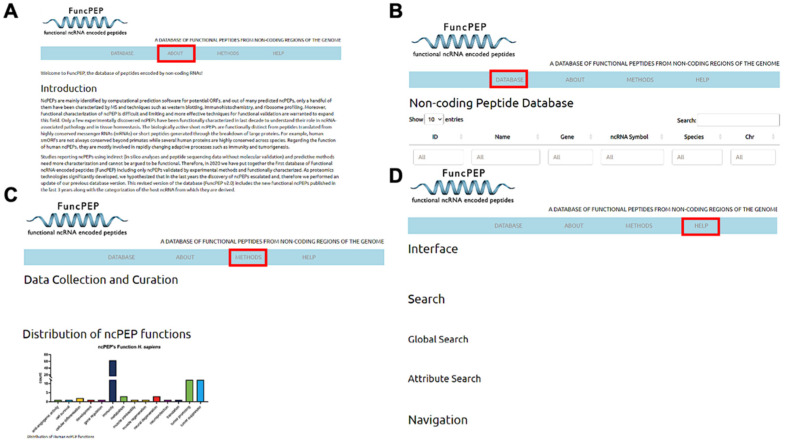
Database interface, as previously described. (**A**–**D**) The wide array of tabs the user can access to find pertinent information regarding the functional peptides included in our database: about, index, info, and help, respectively. The red boxes indicate the pertinent tab being observed for the corresponding panel.

**Figure 3 ncrna-10-00020-f003:**
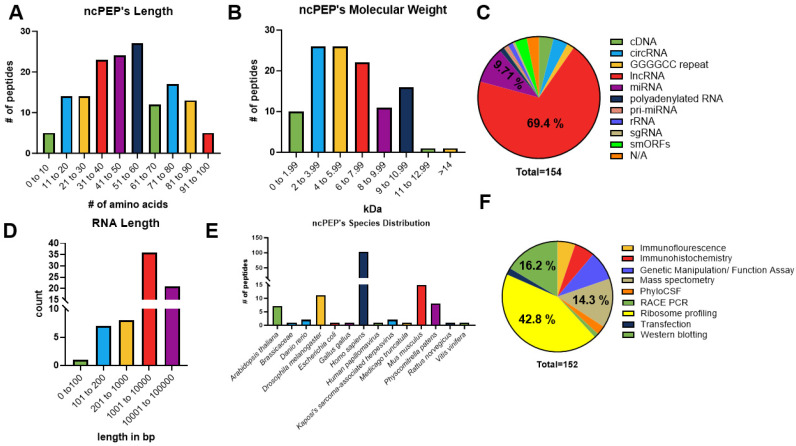
The characterization of ncPEPs included in the FuncPEP database. (**A**) ncPEPs’ length displayed in amino acids. (**B**) ncPEPs’ molecular weight represented as (kDa). (**C**) The classification of all ncPEPs’ host ncRNAs across all species through a part-to-a-whole analysis. (**D**) Distribution of host ncRNAs’ lengths of the included FuncPEPs in the database. (**E**) The distribution of species for the identification of ncPEPs. (**F**) Identification methods incorporated to detect ncPEPs as displayed by a part-to-a-whole analysis. %: ncPEPs. # means “Number”.

**Figure 4 ncrna-10-00020-f004:**
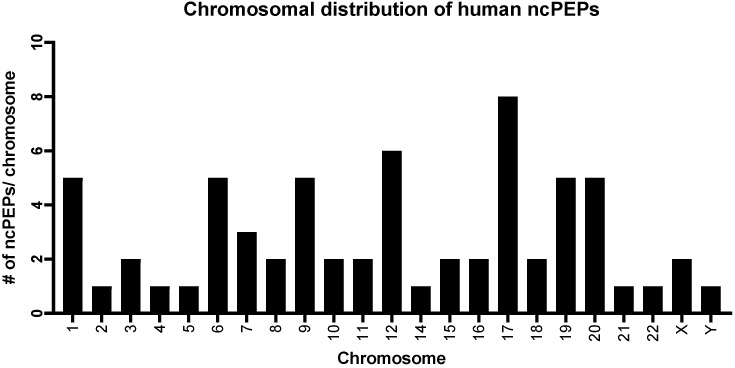
The classification of chromosomal dynamics of the ncPEPs included in the FuncPEP database version 2.0. The chromosomal distribution of human ncPEPs. # means “Number”.

**Figure 5 ncrna-10-00020-f005:**
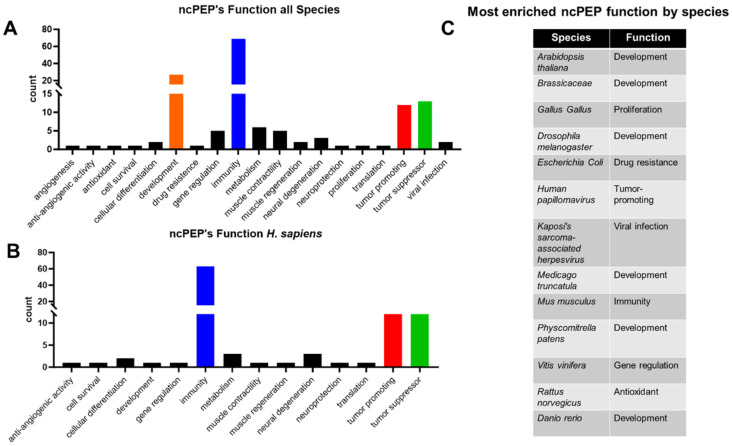
Comprehensive overview of the physiological functions of ncPEPs. (**A**) Distribution of the physiological functions of ncPEPs across all species included in the database. Top enriched functionally relevant physiological pathways are indicated by colored bars. (**B**) Distribution of the physiological functions of human ncPEPs included in the database. Top enriched functionally relevant physiological pathways are indicated by colored bars. (**C**) Table demonstrated the most enriched ncPEP function stratified across each species.

## Data Availability

FuncPEP v2.0 is freely accessible at https://bioinformatics.mdanderson.org/Supplements/FuncPEP/ (accessed on 27 March 2024).

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
