# Peer review of "FuncPEP v2.0: An Updated Database of Functional Short Peptides Translated from Non-Coding RNAs"

_ncrna, 2024, doi:10.3390/ncrna10020020_

Round 1

Reviewer 1 Report

Comments and Suggestions for Authors

The authors present the updated version of databse FuncPEP initially published in 2020. The database contains short peptides translated from non-coding RNA and validated by experiments. The collection is made based on the literature search according to the set of keywords over 3 year period.  The novel version added new functional 93 ncPEPs.

The database is available via the link https://bioinformatics.mdanderson.org/Supplements/FuncPep

I think that the database is important resource and should be published. I have one suggestion to calrify how the functional screen is done? Is this just a summary of have been reported in the original publication? Then it should be stated explicitly. Because now it is unclear what is the method:  “To decipher the role of the retrieved funcPEPs, we performed a functional 209 screen of the candidate ncPEPs included in v2.0 database across all species and then Homo 210 sapiens separately (Figure 5A and 5B). Interestingly, immunity was the most enriched 211 function across all species, as well as for Homo sapiens. 

Author Response

Reviewer 1: The authors present the updated version of databse FuncPEP initially published in 2020. The database contains short peptides translated from non-coding RNA and validated by experiments. The collection is made based on the literature search according to the set of keywords over 3-year period. The novel version added new functional 93 ncPEPs. The database is available via the link https://bioinformatics.mdanderson.org/Supplements/FuncPep I think that the database is important resource and should be published. I have one suggestion to calrify how the functional screen is done? Is this just a summary of have been reported in the original publication? Then it should be stated explicitly. Because now it is unclear what is the method: “To decipher the role of the retrieved funcPEPs, we performed a functional 209 screen of the candidate ncPEPs included in v2.0 database across all species and then Homo 210 sapiens separately (Figure 5A and 5B). Interestingly, immunity was the most enriched 211 function across all species, as well as for Homo sapiens. “
Thank you for the insight comments and compliments towards our manuscript. We also believe that this database is an important resource for publication. We were also interested in that the predominant functional pathways enriched was immunity across all species and homo sapiens. As for clarification, the purpose of this study was to summarize and perform a meta-analysis of the reported functional peptides encoded within ncRNA regions reported in the original publication included in the database. We believe this study this important and serves a meaningful avenue for researchers to standardize and stratify the identified ncPEPs, searched from our database, toward more pre-clinical and clinical studies.
We have changed the following sentence at line: 209 to:
“To decipher the role of the retrieved funcPEPs, we performed a functional summary and meta-analysis of the candidate ncPEPs included in v2.0 database across all species and then Homo sapiens separately (Figure 5A and 5B).”

Reviewer 2 Report

Comments and Suggestions for Authors

The manuscript by Mohapatra et al., titled “FuncPEP v2.0: An Updated Database of Functional Short Peptides Translated from Non-Coding-RNAs” describes the updated version of the FuncPEP database for short peptides (ncPEPs) which are translated from RNAs that were previously annotated as non-coding RNAs (ncRNAs). Recent research has shown that different classes of short and long ncRNAs are translated into short peptides (<100 amino acids) and many of these peptides have novel functions in tumorigenesis, development, immunity, metabolism, and other important cellular processes. While in silico prediction of these peptides is easier, validation of these peptides is difficult using experimental methods. Therefore, only a small fraction of all in silico predicted short peptides are experimentally validated using different methods. Future studies of investigating the mechanism of function of ncPEPs will greatly enhance our understanding of their diverse biological roles in organisms. In this direction, an updated database of experimentally validated ncPEPs will greatly assist researchers in further expanding the knowledge on these interesting biomolecules. The current manuscript describes the new peptides included in the updated database, FuncPEP v2.0 from recently published research articles, and provides the details on the peptides, and their encoding ncRNAs. This is a timely publication, since there is strong interest in ncPEPs and also many novel short peptides encoded from uORFs and other alternate reading frames in mRNAs. The manuscript is concise, well written and discusses the process of selecting experimentally validated ncPEPs, their properties, and their physiological functions. The discussion also talks about their future directions and provides an interesting example of miPEP155, which plays role in antigen presentation by dendritic cells., which will help the readers will appreciate the functions of ncPEPs. Some minor changes the authors need to make are listed below.

Minor issues:

1.     Some of the scientific names of species (Homo sapiens, Mus muscules, etc.) are not in italics throughout the manuscript.

2.     Line 25: Correct deep to deepen. ‘will deepen our understanding..’

3.     Line 61: The length/size of RNAs is indicated in ‘nucleotides’ or nt, and not in base-pairs (bp), as RNAs do not form perfect duplexes like dsDNA. Please change 200 bp to 200 nt.

4.     Line 78: in ‘the’ last decade.

5.     Line 233: please correct ‘specie’ to ‘species’

Author Response

Reviewer 2: The manuscript by Mohapatra et al., titled “FuncPEP v2.0: An Updated Database of Functional Short Peptides Translated from Non-Coding-RNAs” describes the updated version of the FuncPEP database for short peptides (ncPEPs) which are translated from RNAs that were previously annotated as non-coding RNAs (ncRNAs). Recent research has shown that different classes of short and long ncRNAs are translated into short peptides (<100 amino acids) and many of these peptides have novel functions in tumorigenesis, development, immunity, metabolism, and other important cellular processes. While in silico prediction of these peptides is easier, validation of these peptides is difficult using experimental methods. Therefore, only a small fraction of all in silico predicted short peptides are experimentally validated using different methods. Future studies of investigating the mechanism of function of ncPEPs will greatly enhance our understanding of their diverse biological roles in organisms. In this direction, an updated database of experimentally validated ncPEPs will greatly assist researchers in
further expanding the knowledge on these interesting biomolecules. The current manuscript describes the new peptides included in the updated database, FuncPEP v2.0 from recently published research articles, and provides the details on the peptides, and their encoding ncRNAs. This is a timely publication, since there is strong interest in ncPEPs and also many novel short peptides encoded from uORFs and other alternate reading frames in mRNAs. The manuscript is concise, well written and discusses the process of selecting experimentally validated ncPEPs, their properties, and their physiological functions. The discussion also talks about their future directions and provides an interesting example of miPEP155, which plays role in antigen presentation by dendritic cells., which will help the readers will appreciate the functions of ncPEPs. Some minor changes the authors need to make are listed below.
Thank you for the insight comments and compliments towards our manuscript. We also believe that this database is an important resource for publication. We believe this study this important and serves a meaningful avenue for researchers to standardize and stratify the identified ncPEPs, searched from our database, toward more pre-clinical and clinical studies.
Minor issues:
1. Some of the scientific names of species (Homo sapiens, Mus muscules, etc.) are not in italics throughout the manuscript.
Thank you for pointing this out, we have made the appropriate corrections. We italicized all the scientific names.
2. Line 25: Correct deep to deepen. ‘will deepen our understanding..’
Thank you for pointing this out, we have made the appropriate corrections. We changed deep to deepen
3. Line 61: The length/size of RNAs is indicated in ‘nucleotides’ or nt, and not in base-pairs (bp), as RNAs do not form perfect duplexes like dsDNA. Please change 200 bp to 200 nt.
Thank you for pointing this out, we have made the appropriate corrections. We changed bp to nt and defined it.
4. Line 78: in ‘the’ last decade.
Thank you for pointing this out, we have made the appropriate corrections. We added “the” before last decade.
5. Line 233: please correct ‘specie’ to ‘species’
Thank you for pointing this out, we have made the appropriate corrections. We changed “specie” to “species”